# The Fracture Liaison Service of the Virgen Macarena University Hospital Reduces the Gap in the Management of Osteoporosis, Particularly in Men. It Meets the International Osteoporosis Foundation Quality Standards

**DOI:** 10.3390/jcm10184220

**Published:** 2021-09-17

**Authors:** Francisco-Jesús Olmo-Montes, Blanca Hernández-Cruz, Mª José Miranda, Mª Dolores Jimenez-Moreno, Mª Ángeles Vázquez-Gámez, Mercè Giner, Miguel-Angel Colmenero, José Javier Pérez-Venegas, María-José Montoya-García

**Affiliations:** 1Internal Medicine Department, Virgen Macarena University Hospital, Dr Fedriani Avenue No 3, 41009 Seville, Spain; franciscoj.olmo.sspa@juntadeandalucia.es (F.-J.O.-M.); m.j.mir@telefonica.net (M.J.M.); Mdjimenezm@gmail.com (M.D.J.-M.); mangel.colmenero.sspa@juntadeandalucia.es (M.-A.C.); 2Rheumatology Department, Virgen Macarena University Hospital, Dr Fedriani Avenue No 3, 41009 Seville, Spain; perez.venegas@gmail.com; 3PAIDI Research Group CTS/211, Medicine Department, School of Medicine, Seville University, Sanchez Pizjuan SN, 41009 Seville, Spain; mavazquez@us.es (M.Á.V.-G.); merce_giner@yahoo.es (M.G.); pmontoya@us.es (M.-J.M.-G.)

**Keywords:** osteoporosis treatment, quality standards, fracture liaison service type a, gender disparities in OP

## Abstract

Objectives: To describe the Fracture Liaison Service (FLS), to know the characteristics of the patients attended with emphasis on sex differences, and to know the compliance of International Osteoporosis Foundation (IOF) quality standards. Methods: Observational, prospective research. All the consecutive patients that attended in usual clinical practice from May 2018 to October 2019, were over 50 years, and with a fragility fracture (FF), were included. Results: Our FLS is a type A multidisciplinary unit. We included 410 patients, 80% women. FF recorded in 328 women were: Hip (132, 40%), Clinical Vertebral (81, 25%) and No hip No vertebral (115, 35%). Those in 82 men were: Hip (53, 66%), Clinical Vertebral (20, 24%) and No hip No vertebral (9, 10%), *p* = 0.0001. Men had more secondary osteoporosis (OP). The most remarkable result was the low percentage of patients with OP receiving treatment and the differences between sex. Forty-nine (16%) women versus nine (7%) men had received it at some point in their lives, *p* = 0.04. The probability of a man not receiving prior treatment was 2.5 (95%CI 1.01–6.51); *p* = 0.04, and after the FF was 0.64 (0.38–1.09). Treatment adherence in the first year after the FLS was 96% in both sexes. The completion of IOF quality standards was bad for patient identification and reference time. It was poor for initial OP screening standard and good for the remaining ten indicators. Conclusions: the FLS narrowed the gap in diagnosis, treatment, and follow-up of fragility fracture patients, especially men. The FLS meets the IOF quality standards.

## 1. Introduction

Osteoporosis (OP) is a global public health problem [1,2,3]. It causes 8.9 million fractures annually worldwide. It is estimated that 1000 bone fragility fractures (FF) occur every hour. The probability of having a major FF from the age of 50 for the rest of life is 46% in women and 22% in men [1]. Spain is a country with a mean incidence of OP, with 150 to 250 new cases a year per 100,000 adults over 50 years [1,2].

In recent years, there have been advances in the management of patients with OP. Highlights include tools for estimating fracture risk, effective and safe treatments [1,2,3], and incorporating the “Treat to Target” strategy into the disease [4]. Despite this, a gap between OP recommendations and clinical practice continues to be identified [1,2,3]. This large gap occurs in: (i) Identifying cases, (ii) Diagnostic, (iii) Estimating the risk of FF and falls, (iv) Treatment, and (v) Therapeutic adherence [1,2,3]. As an example, after the first hip, 80% of patients do not receive evaluation and/or OP treatment in the following year [1,2,3]. This gap is worse among men [2].

Another overwhelming fact was the decline in the bisphosphonates treatment rate of patients with a hip FF in the United States that happened between 2002 and 2011. That was a drop from 40% to 20%, in the year after the discharge [5]. In these patients the risk of non-treatment among men versus women was 54% higher, *p* < 0.0001 [5]. For most Immune-Mediated Diseases the gender minority falls to women. Among all the patients with OP the worst assessed, diagnosed, and treated are men [2,5,6]. In addition, men have higher rates of secondary OP and a greater treatment gap [5,6]. Over the past 20 years, men have seen an increase in the incidence of FF, particularly in those over 75 years and with hip fractures. In women, the tendency of the FF is to decrease [7].

To narrow the gap and to improve the diagnosis and treatment of patients with OP, Fracture Liaison Services (FLS) were created. The first FLS appeared in the late 1990s in Glasgow, United Kingdom [8]. Subsequently, the “National Institute for Health and Care Excellence” approved its efficiency [9]. Over the years, the FLSs have been consolidated and spread throughout the world as an initiative of the International Osteoporosis Foundation (IOF) [10] and the American Society for Bone and Mineral Research (ASBMR) [11]. In 2017, the European League against Rheumatisms (EULAR) and the European Federation of National Associations of Orthopaedics and Traumatology (ESCEO) joined the initiative [12].

There are currently more than 300 FLSs spread around the world. They work with different models, adapted to the circumstances of each country and health system [1,2,3,8,9,10,11,12,13,14,15,16]. A great heterogeneity is recognised in the different FLS models [12,13,14]. To consolidate these models, Ganda et al. proposed to classify them according to the intensity of the intervention [15], A: There is a coordinator in charge of high intensity or type A. For these, after the first FF the patient is captured, evaluated, and diagnosed. Plans of follow-up and treatment are established together with primary care (PC). B: Type B can have a coordinator, the patient is received and evaluated, and the treatment and follow-up plans are carried out by the PC physician, preferably in conjunction with the FLS. C: In type C, the patient is received and assessed in the FLS. An estimate of the risk of FF is made and recommendations are issued, keeping the patient informed. In this case, the patient care is delegated to the PC physician and the intervention is of lower intensity. D: The type D is the least intensive. It informs the patient, but no evaluation is carried out and follow-up is done by the PC physician, with whom there is no communication [8,9,10,11,12,13,14,15,16].

European guides recommend the FLSs as the secondary prevention standard for FF, although some perform primary prevention [1,2,3,8,9,10,11,12,13,14,15,16]. The ones that work best are those of Type A, in which the risk of FF and falls is estimated, and the therapeutic strategy is shared with PC and are multidisciplinary. That means there is a coordinator who organises orthopaedic surgery and traumatology, rehabilitation, rheumatology, and “fall prevention services” [8,9,10,11,12,13,14,15]. Regardless of which FLS model is implemented, the IOF, and the National Osteoporosis Foundation (NOF) set the quality standards that they must meet [8,9,10,11,12,13,14,15].

These standards called “The Best Practice Framework” have already been published [13,14,15]. A standard compliant FLS has demonstrated numerous benefits. Reducing between 15% and 30% in re-fracture rates. Reduction in hip fracture mortality after the first month of 0.73 (CI95% 0.65–0.82); and after the first year of 0.81 (0.75–0.87) [17]. Besides, a significant increased risk estimation of FF, falls and evaluation of secondary causes of OP, lead to an increase of prescription from 15% to 40% [8,9,10,11,12,13,14,15,16]. The Virgen Macarena University Hospital (VMUH) transformed its FLS (VMUH-FLS) in May 2018 into a multidisciplinary unit, with improved communication with PC [18]. We decided to make an initial evaluation after 18 months of operation. The aims of the study were: (i) To describe the characteristics of the FLS and how it works, (ii) To know the type of patients being treated, with an emphasis on gender differences, and (iii) To know the degree of completion of The Best Practice Framework quality standards according to IOF.

## 2. Materials and Methods

### 2.1. Study Design

This study was a a prospective, observational, and analytical one in regular clinical practice. It took place from 1st May 2018, to 31st October 2019. Setting: VMUH is an 850 bed, teaching hospital, serving a population of approximately 481,296 inhabitants (157,428 over 50 years old) in Seville, Spain. This population can be treated in 34 PC health areas [18]. The VMUH belongs to the Public Health System of Andalusia, with universal coverage for medical consultation, laboratory tests, X-Ray films, bone densitometry test (BMD). It also covers all specific treatments for OP. All candidate patients are valued for inclusion in VMUH-FLS. After the signing of the informed consent, they are consulted face to face. Their clinical data are included in a real time database of 106 variables in a specialized free software for electronic case report form (OpenClinica^®^ LLC Enterprises Services Group, USA) [19], as described in the results section.

### 2.2. Inclusion and Exclusion Criteria

We included patients of ≥50 years old, cared for in the VMUH-FLS for an FF occurring in the previous 24 months. They agreed to participate in the study and gave their informed consent. FF was defined as a confirmed X-ray bone fracture, which derives from trauma that under normal conditions does not cause a fracture, after a fall from the patient’s height and without acceleration mechanism. In the case of having two or more FFs, the index was coded with the greatest impact on physical function, quality of life, and mortality (hip, vertebra, humerus, distal end of radius (DER), and any others in the same order).

Exclusion criteria: (i) Patients of under 50 years of age, (ii) Patients who did not agree to participate, (iii) Patients with a noticeably short life expectancy (defined by a Paliar Index ≥10 points with a mortality risk at six months >61%) [20], and/or iv. Patients with trauma or pathological fractures.

### 2.3. Statistical Analysis

In a first phase descriptive statistics were calculated with measures of central tendency and dispersion. After that, a graphic analysis was conducted, comparing the main variables by sex, age in decades, and by FF type. Finally, parametric and non-parametric tests were calculated, according to the type of variable. Both sexes were at this point compared, considering as null hypothesis of no differences between them. A two-tailed *p* < 0.05, with Bonferroni correction for multiple comparisons was considered significant. The statistical analysis was carried out with the STATA v 10.1 package [21].

Sample size calculations. Descriptive analysis does not require sample size calculations. All patients included in the chosen period were selected and analysed.

## 3. Results

### 3.1. VMUH-FLS Characteristics and Composition

Figure 1 shows both characteristics and composition. They are described below.

#### 3.1.1. Staff and Patients

The staff is composed by a secretary, two specialists in internal medicine, one rheumatologist, one case management nurse, two densitometer technicians, and three PhDs from the research group of the Andalusian Research Plan (PAIDI CTS/211). The clinical coordinator is an internist physician (FJOM), who relies on the case management nurse (MDJM). The head of the group is an internist and professor at the Faculty of Medicine (MJMG).

The type of patients attended are hospitalised and outpatients.

#### 3.1.2. Pathways of Referral

The pathways are multiple, and a direct derivation from PC and various referral services serving hospitalised and outpatients is important. In addition, index cases are identified by the reviewing of emergency lists looking for codes related to FF according to the Minimum Basic Data Set of Andalusia (CMBD) [22] and the International Statistical Classification of Diseases and Related Health Problems 10th Revision (ICD-10) [23].

Regular visits are carried out to the Plaster and Emergency Room with review of outpatient lists. Surgical demand records are searched for admitted patients. There is also direct derivation by consultation sheets of the reference units (Orthopaedic Surgery and Traumatology, Rheumatology, Internal Medicine, Rehabilitation, PC, and others such as Medical and Radiotherapy Oncology, Urology, Gynaecology, and Endocrinology).

#### 3.1.3. Flow of Patients

There is a dual flow of patients to the unit:Two-way process: The collaboration with PC is critical and involves sending patients from PC, where they are cared for with an FF. Once evaluated in VMUH-FLS, patients are referred to PC for follow-up, with the option to be re-referred to the VMUH-FLS, if needed. From the reference units, patients with an FF are referred to the VMUH-FLS and vice versa, based on an early care agreement;One-way process: patients with an FF are derived from referral services, served in the VMUH-FLS, and referred to PC (Figure 1).

#### 3.1.4. Referral Protocols

Previous working meetings were held with all those concerned. They agreed to centralise care in VMUH-FLS. The corresponding diagnostic and treatment protocols were agreed, based on the SEIOMM Clinical Practice Guidelines [24] and the Spanish Society of Rheumatology (SER) [25]. Signage was placed with referral instructions in Traumatology and Orthopaedics consultations, Emergency rooms, and Plaster rooms. In these areas pre-printed inter consultations and lab test sheets were provided. Well-functioning pathways were established by telephone, e-mail, WhatsApp, and Tele-consultation.

The referral protocols are different depending on the type of FF and are outlined in Figure 1.

Hip fracture: The patient is admitted in the emergency room service and is transferred to the Traumatology and Orthopaedics Units. They contact the Perioperative Internal Medicine Unit that proceeds with the ortho geriatric evaluation. When discharged, the patient is referred to the FLS;DRE fracture: Most are served in the Emergency room and move to the plaster room, where they are evaluated and derived to the FLS. Those requiring surgical treatment are captured by CMBD [22] or ICD-10 [23];Humerus fracture. The patient is admitted through the Emergency Room. If required, they are hospitalised in Orthopaedic Surgery and Traumatology Unit. When discharged, they are derived from consultation of trauma and from there to the FLS;Vertebral fractures. They are captured and forwarded from all reference units and from PC.

#### 3.1.5. FLS Care

It is carried out on a one-time visit that lasts approximately 3.5 h. In a day of consultation, a physician attends to 12 first visits and four reviews. There are two consultations on Tuesdays and two on Fridays. The rest is by telemedicine.

Once a possible case has been identified, the nurse checks the medical history to verify that the patient meets the inclusion criteria. In this case, she contacts the patient by phone and fixes an appointment, during which, and after explaining the project and signing the consent, a physician and nurse carry out the comprehensive evaluation.

The evaluation includes (i) Anamnesis, (ii) Physical examination with anthropometric measurements (weight, height, and body mass index (BMI)), (iii) Collecting sociodemographic variables, iv. Identifying the mechanism that caused the fracture, smoking, alcoholism, and personal history of fractures and history of falls and treatments, v. Estimating the risk of fracture according to the instrument Fracture Risk Assessment Tool (FRAX^®^) [26].

In addition, some validated Spanish versions of the following specific tests have been used:Barthel scale [27];Hand grip strength by dynamometry [28];Fall risk estimation with J D Dowton Index [29];Functional performance with the Short Physical Performance Battery Scale (SPPB) [30];Screening of the nutritional status with the Mini Nutritional Assessment instrument (MNA instrument ^®^) [31];Generic quality of life measurement (EuroQol5D) [32].

Physicians review the results of a specific analysis: full blood count, Erythrosedimentation rate (ESR), serum levels of calcium, phosphorus, alkaline phosphatase, parathormone, 25(OH) vitamin D, total proteins, albumin, creatinine, glomerular filtration rate, thyrotropin, and tyrosine, basic urine analysis, and calcium/creatinine ratio in urine.

Other tests are added according to clinical guidance (antibodies to transglutaminase and gliadin, protein electrophoresis, testosterone levels, and 24-h urine analysis or autoantibody tests).

The search for morphometric vertebral FF is performed by reviewing lateral X-rays of the dorsal and lumbar spine, with a specific protocol agreed with the Radiology department. The femur neck and lumbar BDM with a dual energy X-Ray absorptiometry (Dexa Hologic Discovery densitometer^®^) is performed and assessed.

Comorbidity and the type and number of drugs at the time of FF are evaluated. Looking for drugs (glucocorticoids, aromatase inhibitors, therapy of prostate cancer, psychotropics, etc) related to OP and falls is important.

Clinical evaluation is recorded in a real time database (OpenClinica^®^) containing 106 variables [20]. A report is obtained from the database and is computerised in the Andalusian Health Service (Diraya) [33]_._ With all the information the patient is re-evaluated in a single act, during which on suspicion of sarcopenia, femoral rectum muscle ultrasound is performed (ultrasound General Electric Logiq v2^®^).

Finally, a diagnosis is made regarding OP, risk of falls, FF and a comprehensive targeted treatment plan is done for each patient. The plan includes:Verbal and written explanation of OP;Measures of primary prevention of falls and OP;Nutrition indications;Exercise plan to improve physical function;A plan to reduce the risk of falls.

A written report is issued, including a table of exercises aimed at improving muscle balance and function, and a specific drug treatment for OP is instructed. A copy of the report is forwarded to the PC physician.

Follow-up. It is done based on risk. Patients at low and medium risk of FF and falls are referred to PC and monitoring is conducted by phone in the following 6, 12, and 24 months.

Patients at high risk of FF and falls, and those with suspected poor adherence to treatment are followed in FLS with personal appointments every 3, 6, 12, and 24 months, depending on the judgment of the physician.

Adherence is measured by phone or face-to-face interview; and confirmed by reviewing the dispensing of the drug through electronic prescription of the Andalusian System of Health [34].

#### 3.1.6. Patient Recruitment

A total of 450 patients were included in an 18-month study. Of which 351 (78%) were attended to in single-act consultation. The patient flowchart is shown in Figure 2.

Forty patients (9%) were excluded. The causes of exclusion were a noticeably short life expectancy (Paliar index ≥10) in 12 (3%) patients, six patients (1%) with age <50 years, three patients (0.6%) with high-impact fractures, four patients (0.9%) with refusal to participate. In total, 15 duplicated records were removed after a database clean-up.

### 3.2. Clinical Characteristics and Treatment of the Patients

Data from 410 patients were analysed and are shown in Table 1, aged 73.5 (±10.2) with a lower limit of 51 and upper limit of 94 years. Most were women (*n* = 328, 80%).

There were no differences between women and men in terms of age, referral time, or BMI, but in weight and height. Men were taller and heavier, as expected. There were clinical and statistical differences in the type of FF (Table 1 and Figure 3).

Hip FF was more common in men, whereas DRE and humerus was more common in women. Vertebral FF and the Other FF group had equal gender distribution. In 11% of cases two or more FF were found. This percentage ranged from 5% to 12% in patients with first hip, vertebral, and DRE FF and rose to 67% in the group with other FFs. The anatomic sites of the second FF were vertebral *n* = 9 (21%), DRE *n* = 6 (14%), tibia *n* = 6 (14%), and+ fibula *n* = 4 (9%).

Two-thirds of patients were evaluated in the FLS within 6 months after the index FF. Most of the FF occurred in patients between 70 to 79 years of age, followed by patients of age between 80 to 89 years, with no gender differences. The prevalence of smoking was three times higher in men. The prevalence of alcoholism was five times higher. Additionally, secondary causes of OP were commonly found in men. In addition, they had lower physical activity. There were differences in the referral units to FLS, related to type of FF.

#### Sex Differences

The most important gender difference was the low percentage of patients receiving OP specific treatment prior to FF; 49 (16%) women versus 9 (7%) men had been treated at some point in their life before the FF, *p* = 0.04. After the FF, specific treatment for OP was started in the reference unit in 271 (86%) women versus 48 (63%) men (Table 1 and Figure 4).

All patients, both women and men were treated after the VMUH-FLS intervention. The probability for a man not to receive treatment at some point prior to FF was 2.5 (CI95% 1.01–6.51); this difference lowered to 0.64 (0.28–1.09) in the referral unit after the FF; and disappeared in the FLS, as 100% of patients of both sexes received specific treatment.

The drugs used are shown in Figure 5. In total, 65% started treatment with bisphosphonates; 24% with denosumab; 8% with teriparatide and the rest selective oestrogen receptor modulator receptors or hormone replacement therapy.

In 188 (47%) patients the specific treatment for OP started in the referral unit was changed in the VMUH-FLS, after identifying barriers in therapeutic adherence. The most common change was from oral bisphosphonate to zoledronic acid IV in 20% of the cases.

### 3.3. VMUH- FLS Compliance with the IOF Quality Standards

Figure 6 summarises the data with compliance with the quality standards.

The completion of IOF quality standards was poor (red traffic light) for patient identification and FLS reference time items. It was average (amber traffic light) for initial OP screening standard and was good (green light) for the remaining 10 indicators.

Quality standard No 1. Patient identification. The review of the hospital’s computerised register by ICD-10 codes in 2019, reported that 1008 patients over 50 years were treated for FF in 18 months. In the same period 450 candidate patients were identified, and 410 were evaluated and included in the database. Only 40.4% of potential candidates had been identified. This item received a red traffic light. Of those identified patients, 91% were included. The frequency with which patients declined to participate was only a 0.09% (Figure 2);Quality standard 2. Vertebral fracture assessment. Targeted search for vertebral FF was conducted to 389 (96%) patients using X-rays. This standard received a green traffic light. In total, 101 (25%) of these patients had suffered clinical vertebral FFs. Another 79 cases (19%) had morphometric vertebral fractures. 52 (66%) morphometric fractures in patients with hip FF; 12 (15%) in patients with DRE, 10 (13%) with humerus FF, and 5 (6%) in patients with other types of FF, *p* = 0.0001.Quality standard 3. Time lapse from FF to VMUH-FLS assessment. The evaluation in the FLS was done after 5.9 (±4.7) months, with a lower limit of 3 days and an upper limit of 24 months. Nearly one-third (27%) were evaluated in the following three months after the FF; and 71% within 6 months after the FF. The indicator received a red traffic light.Quality standard 4. Adherence to guidelines and number of patients with DXA. Each month the members of the FLS meet and discuss adherence to the guidelines, especially in those difficult or refractory cases. Regular meetings with the reference units and PC were also held. Thus, treatment guidelines are known and applied in a 98% of cases. As a sample, a DMO was prescribed and performed to 365 (89%) patients in the femoral neck and to 366 (90%) in the lumbar spine. The indicator received a green traffic light.Quality standard 5. Fall risk estimation and prevention. The risk of falls was valued with the SPPB test to 298 (73%) and the Downton scale to 339 (83%) of the patients. In all cases where the risk of falls was high, the nurse initiated an oral and written exercise programming, that was supported by the physician. The indicator received a green light.Quality standard 6. Secondary causes of OP screening. A cause of secondary OP was sought in all patients by anamnesis and physical examination. The results of the blood and urine tests aimed at investigating secondary OP causes were collected in the database in 61% to 77% of cases, as shown in Figure 6. The indicator received amber light.Quality standard 7. Multidimensional assessment of potentially modifiable health and lifestyle. Estimations of level of regular exercise and of the degree of physical dependence; the use of OP related drugs, the use of fall related drugs and dairy consumption were made in 92% of the cases. The indicator received a green light.Quality standard 8. Specific treatment for OP. Following the evaluation in FLS, specific treatment for OP was initiated in 100% of cases. The indicator received a green light.Quality standard 9. Adherence of specific treatment for OP at 12 months. The nurse contacted all patients by phone at 3, 6, and 12 months. The adherence at 12 months was 95%. This information was compared with the review of the dispensing of the drug in the electronic prescription and the fulfilment at 12 months was 94%. The indicator received a green light.Quality standard 10. Communication strategy. In addition to verbal information, clinical reports were issued. The clinical reports included: (i) The results of the FLS evaluation, (ii) The risk of FF, (iii) The risk of falls, iv. The treatment strategy, with both non pharmacologic and specific OP drugs. These reports were handed over to each patient and a copy was sent by post to all PC physicians in 99% cases. The indicator received a green light.Quality standard 11. Database recorded. Data of each patient were recorded in real time for 98% of the patients. The level of completion of the variables was 80%. The indicator received a green light.

## 4. Discussion

This observational and prospective study analyses the first 18 months of operation of an FLS at a teaching hospital in the Public Health system in southern Spain. The VMUH-FLS treats external and in-patients after an FF. The aim of the study was to describe the characteristics and the operation of the VMUH-FLS, to know the type of patients seen after 18 months of operation, and to know the level of completion of the quality standards according to the IOF and NOS.

It is a Type A and multidisciplinary FLS. In this we proactively intend to: (i) Identify patients accurately with a new FF, including vertebral ones. (ii) Assess and stratify the FF risk, fall risk and severity of OP. (iii) Treat each case with no pharmacological measures and of course with specific drugs for OP in the long term. iv. The follow-up of the patients. [8,13,14,15,16].

Similar units are described in the United Kingdom [8,9], Canada [35], Sweden [36], The Netherlands [37], France [38], Italy [39], Spain [40], Australia [41]. There are models in development in the Asia-Pacific region [42]. FLSs fit one of the four models described, and all of them demonstrate advantages over the standard clinical practice [8,9,10,13,14,15,16,17,34,35,36,37,38,39,40,41]. Its implementation and sustainability require the organization of a multidisciplinary team and a lot of work and commitment of the participants.

The figure of the clinical leader to reach agreements and collaboration with the reference units and PC is key since these referral units are different from each other. For example, surgical versus medical, emergence units and elective care, and PC [8,9,10,11,12,13,14,15]. This clinical leader seeks common protocols for identifying, diagnosing, treating, and monitoring patients agreed with all these referral units. In VMUH-FLS this is achieved through sessions and narrow communication channels, commented in the FLS description.

The second key figure is that of the nurse, who coordinates the functioning of the unit and the patient care. She is essential in risk estimations, patient follow-up, and treatment adherence. She is the connection among the reference units, PC, and the patient [8,9,10,11,12,13,14,15].

The third pillar of the FLS is the research group with its leader. They have made it easy to connect between the healthcare and the research. Sustainability is guaranteed as our unit gets its funds from resources of the Internal Medicine and Rheumatology Departments, use their own resources from the Andalusian Health System. All the staff spend a day a week within their usual clinical practice. However, ensuring the funding of the nurse is an outstanding issue, as it should not depend on scholarships, but should also be staff of the Health Service [8,9,10,11]. We hope that when we have results of health indicators, such as reduction in re-fracture and mortality rates, the nurse will become a full-time worker.

When analysing the characteristics of all the 408 patients, these patients are like those with OP in Europe [1,2,3,8,9,13,14,15,16,17,35,36,37,38,39] and Spain [40]. In Europe, one in two women and one in three men of over 50 years will have an FF, and Spain is one of the European countries with one of the most ageing populations in the world [10,11,12,13,14,15,16,17,18,19,20,21,22,23,24,25,26,27,28,29,30,31,32,33,34,35,36,37,38,39]. More and more fractures are being addressed in patients >80 years, and the differences between gender decrease in them [1,2,3,7]. A recent systematic review of 33 FLSs shows great heterogeneity in the type of patients and FF, depending on the individual characteristics of the FLSs and the health system where they are based [43]. This review showed patients between the age of 64 and 80, with a man rate of 13% to 30%, and only two FLSs included women. Our study involved four women (80%) for each man (20%), in the upper range of the distribution.

In the review as well as in our study, differences in the type of FF between genders were found. Hip FF occurred most often in men, spine FF alike in both men and women, and DER in women. Men most often had secondary causes of OP, smoking and alcoholism, as well as lower physical activity. These data are described in the literature [1,2,3,7,8,9,13,14,15,16,17,36,37,38,39,40,41]. As far as 2013, Ganda and colleagues had noticed gender disparities in FLS, in the identification, research and treatment of OP [15].

Our data show that men with FF exercise less than women, perhaps because they have more smoking and drinking habits. In our study the most significant difference was the low proportion of men (7%) in contrast with women (16%), who received specific treatment for OP prior to FF and resembling those found in a previous report [7]. After the attention by the FF index in the reference unit, it increased to a 37% and a 47%, respectively. And in the VMUH-FLS it was of 100%, for both sexes. Undoubtedly the implementation of the FLS has helped to improve treatment rates, not only in the FLS per se, but in the reference units, as described [1,2,3,8,9,10,11,12,13,14,15,16,17].

A second systematic review confirms these differences and disparities between genders in the health care of patients with a first FF. This shows how the disparity tends to improve after the attention at FLS [44]. Not only do men receive specific treatment for OP less frequently. They also see doctors less frequently, and therefore the FLS. Furthermore, they are given fewer diagnostic studies, including BMDs [44,45,46]. Their adherence and therapeutic completion are lower, and they are subjected with higher mortality after a hip FF [1,2,3,7,15,44,45,46]. In this way, the inclusion in FLS decreases significantly gender disparities.

The VMUH-FLS allows the identification of patients of 80 or more years, assuming one third of the total. These elderly people, who often have frail health, constitute a group of “very high” risk of refracture and increased mortality. Some special care must be taken for them, with the collaboration of the orthogeriatric units [47]. These elderly people should be treated in an intensive way, with a specific plan of no pharmacological and pharmacological treatment, and it should be equal between women and men, younger or older without differences.

Quality of care indicators: the initial evaluation of VMUH-FLS performance allows the identification and problem solving and confirming that the gap in the OP is reduced [13,14,15,16]. However, there are some reports that define these quality indicators—transparency in information is scarce [8,9,10,11,12,13,14,15,16,45].

A recent paper has improved this information and added a traffic light system for its evaluation, based on the percentages at which the indicator is reached [48]. After the analysis we find a poor assessment (red light) in two of the items. The first was patient identification, while the computer system, both CMBD [20] and ICD-10 [21], allows us to identify more than 99% of FFs. The problem arises when capturing these patients and including them in the VMUH-FLS. During the first 18 months we captured only 40% of the patients in the FLS. This low rate of patient uptake is multifactorial. We place it on the learning curve of all the members of the unit. Another possible cause is the lack of interaction with the referral units, mainly PC. PC sent us only a 10%of the patients. For this reason, we must improve communication and interaction flows with PC and other referral units, to improve patient recruitment. This communication and the use of the data provided to us by the statistics and computers lists, will improve the recruitment of both out and in-patients.

The other point with a red traffic light was the time elapsed between the FF index and the evaluation of the patients in the VMUH-FLS. 70% of patients were treated at FLS in the first 6 months after FF, and only one-third in the first 3 months, according to recommendations. This period can be improved. Again, the learning curve caused a delay in the FLS attention. The fact that hip FF is common also contributes to this. Hip FF leads to poor outcomes in mobility and physical function. Some patients, of 80 or more years and a hip FF, delayed the appointments to the FLS due to mobility problems. Both the ortho geriatric service and learning of the FLS members, as well as being aware of the failure, will minimise the delay.

The amber light indicator was the assessment indicator. This organizational problem was detected at the outset. The current coordination between secretariat, blood sample-taking service, densitometer rooms and radiology has collaborating in the improvement of this figure.

The advantages of the study are (i) This is a prospective patient cohort, (ii) It includes patients of usual clinical practice in a real-time database, (iii) It incorporates out and in-patients, and iv. The FLS is Type A. We also have resources for its implementation and maintenance. Being the resources mainly, except for the nurses, dependent of the health system, our FLS will be sustainable in the years to come. Another advantage is the multidisciplinary team with efficient leaders, and all the staff performing their work properly.

The limitations of the study are the low percentage of patients enrolled. We expect to increase it over time in the future. Elderly patients with comorbidities, treated with drugs related to OP, reduced physical capacity and cognitive decline, with high risk of falls and re-fractures require concentrated efforts in the fields of Internal Medicine, Rheumatology, Rehabilitation, Orthopaedic Surgery, and Traumatology, which are sometimes complicated.

## 5. Conclusions

HUVM-FLS is a type A multidisciplinary unit that in 18 months of operation has identified, evaluated, and treated 408 patients with OP and an incident FF. Its operation has narrowed the gap in diagnosis, treatment, and follow-up of FF patients, especially men. It is essential to improve patient recruitment, reduce referral times and increase the overall assessment of the patients.

## Figures and Tables

**Figure 1 jcm-10-04220-f001:**
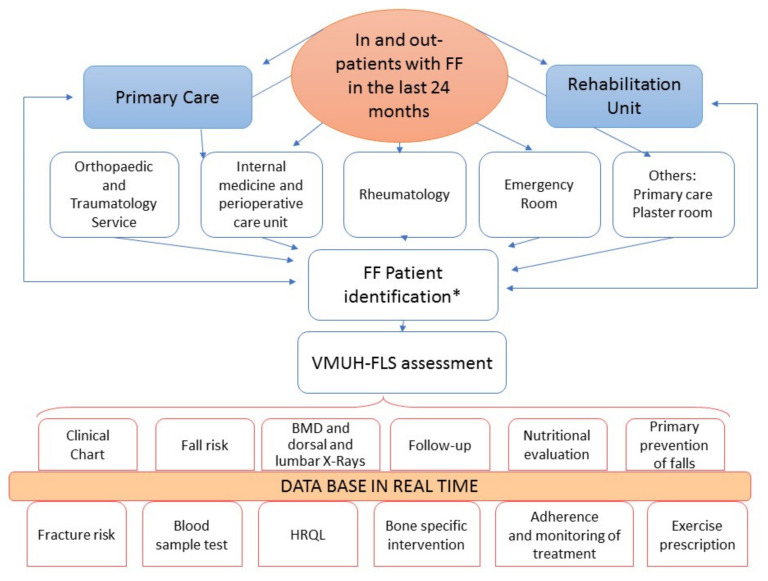
Scheme of the Fracture Liaison Service. VMUH Virgen Macarena University Hospital; FLS Fracture Liaison Service; FF Fragility fractures. * Assessment by the nurse and included if Paliar index ≥10.

**Figure 2 jcm-10-04220-f002:**
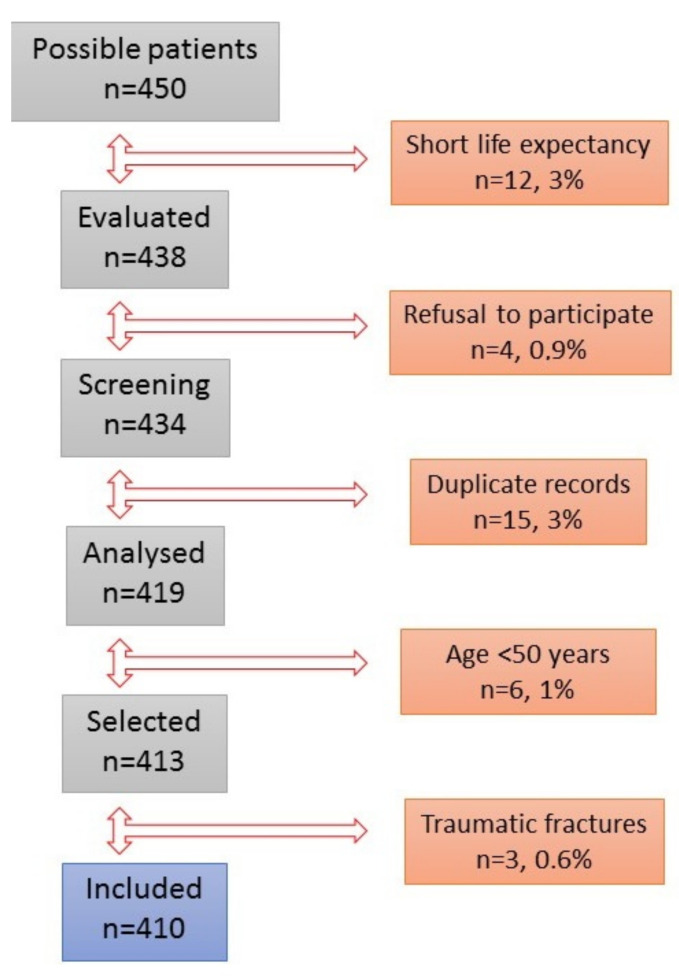
Flow diagram of the 450 patients attended in the Fracture Liaison Service.

**Figure 3 jcm-10-04220-f003:**
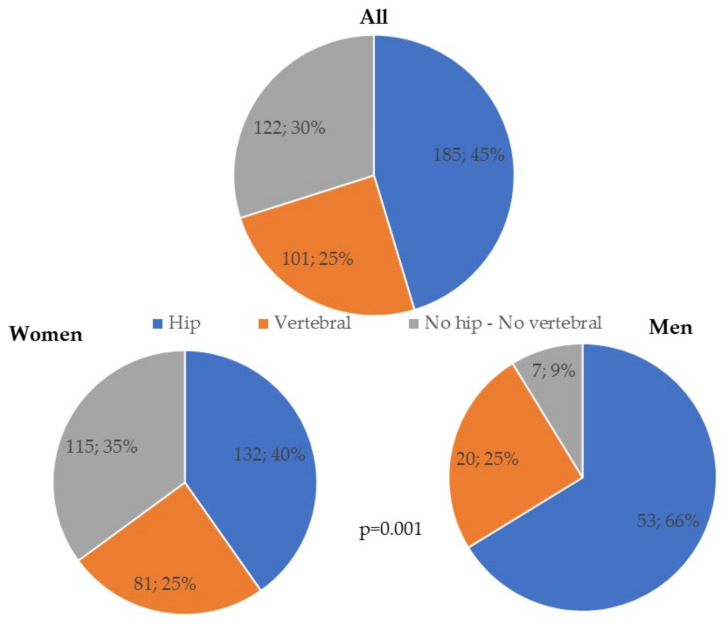
Type of bone fragility fractures in all patients and by sex. No hip-No vertebral fragility fractures include distal radius end *n* = 65 (16%), humerus *n* = 32 (8%), tibia *n* = 5 (1%), fibula *n* = 4 (0.9%), tibia and fibula *n* = 4 (0.9%), ribs *n* = 4 (0.9%), pelvic fracture *n* = 4 (0.9%), elbow *n* = 2 (0.4%), patella *n* = 2 (0.4%) and one distal femur.

**Figure 4 jcm-10-04220-f004:**
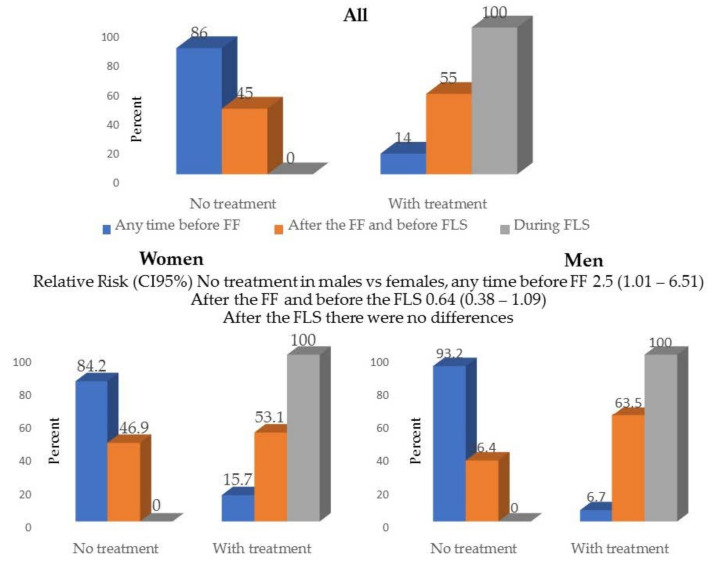
Specific treatment for osteoporosis previous and after the Fracture Liaison Service in all patients and by sex. FF Fragility fracture FLS Fracture Liaison Service.

**Figure 5 jcm-10-04220-f005:**
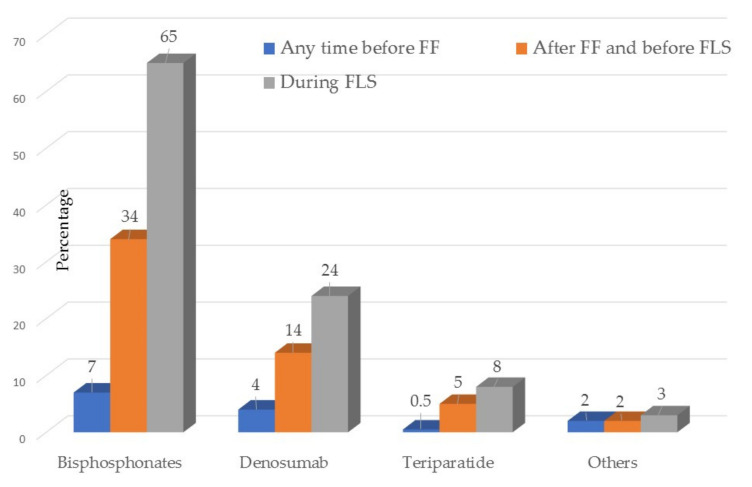
Any time in the past and current specific drugs for osteoporosis. FF Fragility fractures FLS Fracture Liaison Service.

**Figure 6 jcm-10-04220-f006:**
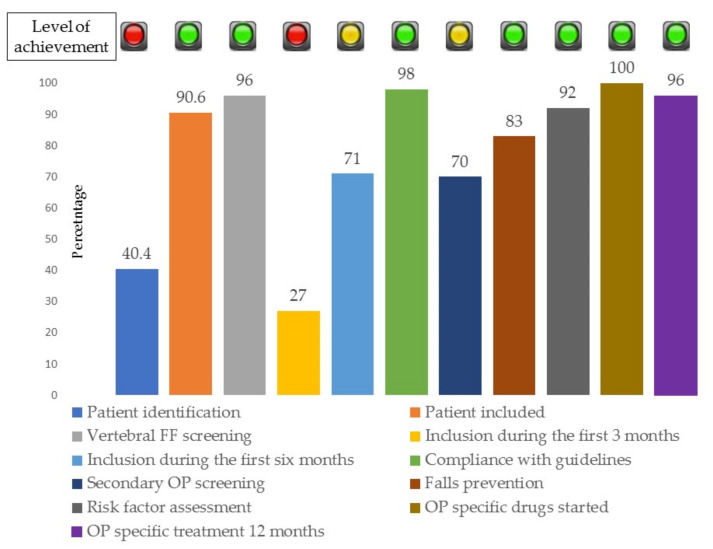
Percentage of completion of the International Osteoporosis Foundation quality standards for the Fracture Liaison Service.

**Table 1 jcm-10-04220-t001:** Characteristics of patients.

Variable	Women	Men	Total	*p* *
	*n* = 328 (80%)	*n* = 82 (20%)	*n* = 410 (100%)	
	Mean ± SD	Mean ± SD	Mean ± SD	
Age, years	73.5 ± 10.3	73.4 ± 10.2	73.5 ± 10.2	0.9
Age of menopause	48.9 ± 5.1			
Months since FF and FLS assessment	5.9 ± 4.7	5.4 ± 4.4	5.8 ± 4.6	0.4
Weight (kg)	67.0 ± 13.7	75.2 ± 14.7	68.6 ± 14.3	0.00001
Height (m)	1.52 ± 0.06	1.64 ± 0.06	1.54 ± 0.08	0.00001
BMI	28.7 ± 5.5	27.8 ± 4.8	28.6 ± 5.4	0.1
T-Score of femoral neck	−2.32 ± 1.11	−2.36 ± 0.82	−2.33 ± 1.06	0.7
T-Score of total lumbar spine	−1.99 ± 1.39	−1.66 ± 1.60	−1.92 ± 1.93	0.1
	*n* (%)	*n* (%)	*n* (%)	
Type of fracture				0.0001
Hip	132 (40)	53 (66)	185 (45)	
Vertebra	81 (25)	20 (24)	101 (25)	
DRE	64 (20)	1 (1)	65 (16)	
Humerus	31 (9)	1 (1)	32 (8)	
Other	20 (6)	7 (8)	25 (6)	
Total	328 (80)	82 (20)	410 (100)	
Time between FF and FLS visit (months)				
0–3	82 (27)	22 (29)	104 (27)	
3–6	130 (43)	37 (48)	167 (44)	
6–12	57 (19)	8, (10)	65 (17)	0.3
≥12	36 (12)	10 (13)	40 (12)	
Total	305 (80)	77 (20)	376 (100)	
Age group (years)				
50–59	47 (15)	12 (15)	59 (15)	0.9
60–69	71 (22)	20 (25)	91 (23)	
70–79	110 (34)	25 (31)	135 (34)	
80–89	85 (27)	22 (27)	107 (27)	
≥90	8 (2)	1 (1)	9 (2)	
Total	321 (80)	80 (20)	401 (100)	
Current smoking	37 (12)	27 (34)	64 (16)	0.0001
Alcoholism >3 units/day	16 (5)	19 (24)	35 (9)	0.0001
Secondary OP	41 (15)	17 (27)	58 (17)	0.02
BMD femoral neck				
Osteoporosis	138 (47)	28 (14)	160 (46)	0.3
Osteopenia	97 (33)	32 (46)	129 (35)	0.03
Normal	60 (20)	9 (13)	69 (19)	0.1
BMD total lumbar spine				
Osteoporosis	120 (40)	24 (35)	144 (39)	0.4
Osteopenia	78 (26)	14 (21)	92 (25)	0.3
Normal	100 (34)	30 (44)	130 (36)	0.1
BMI ≤ 19	9 (3)	0 (0)	9 (2)	0.1
BMI ≥ 30	125 (38)	32 (40)	157 (38)	0.7
Calcium intake in diet				
≤ 1000 mg/day	142 (44)	30 (38)	172 (43)	0.3
>1000 mg/day	180 (56)	44 (62)	229 (57)	
Regular exercise				
Daily exercise	44 (15)	5 (7)	49 (13)	0.0001
Stroll on the street	141 (47)	36 (48)	177 (47)	
Wander at home	87 (29)	15 (15)	102 (27)	
Bed-armchair	29 (10)	19 (25)	48 (13)	
Total	301 (80)	75 (20)	376 (100)	
FLS referral unit				
Orthopaedic surgery and traumatology	134 (41)	27 (34)	161 (40)	
Perioperative internal medicine	101 (31)	39 (49)	139 (34)	
Rheumatology	39 (12)	7 (9)	46 (11)	0.004
Others	38 (12)	7 (9)	45 (11)	
Emergency room and plaster room	5 (3)	0 (0)	5 (3)	
Total	320 (100)	80 (100)	410 (100)	
Pre-FF OP treatment				
None	262 (49)	69 (93)	331 (86)	
Specific treatment	49 (16)	5 (7)	54 (16)	0.02
Total	311 (81)	74 (19)	385 (100)	
Treatment for OP after FF				
None	146 (47)	27 (36)	173 (50)	
Specific treatment	165 (53)	47 (63)	212 (55)	0.1
Total	311 (80)	74 (20)	385 (100)	
Treatment for OP after FLS				
None	0 (0)	0 (0)	0 (0)	
Specific treatment	310 (100)	73 (100)	383 (100)	

OP Osteoporosis, SD Standard deviation, FF Fragility Fracture, FLS Fracture Liaison Service, BMI Body mass index. DRE Distal radius end. BMI Bone mineral densitometry. * Comparison between women and men.

## Data Availability

Per protocol, all the materials related with the study are stored in the FLS. The electronic clinical records of each patient are in their Diraya, a special software of the Andalusian Health System. The database of FLS is filled in real time and is stored in OpenClinica^®^, a special module of the Diraya, on the web page of the Andalusian Health System. The data of the patients are confidential and protected by the current organic data protection law.

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
