# Peer review of "The Fracture Liaison Service of the Virgen Macarena University Hospital Reduces the Gap in the Management of Osteoporosis, Particularly in Men. It Meets the International Osteoporosis Foundation Quality Standards"

_jcm, 2021, doi:10.3390/jcm10184220_

Round 1

Reviewer 1 Report

1. The paper is very readable and thoroughly written. DOI numbers missing, images unreadable.

2. A very interactive study showing the current problems of the world.

3. Original topic, brings innovation a different perspective.

4. The text is clear and easy to read.

5. Coherent conclusions, presented very clearly 

Author Response

Answers to reviewer 1.

I greatly appreciate the kindness in reviewing our work. This makes it better. Without this, the researchers would have many problems in improving.

  1. The paper is very readable and thoroughly written.

Answer BHC: Thank you for the comment.

  • DOI numbers are missing.

Answer BHC: The DOI number is assigned after that the submission is accepted.

  • Images unreadable.

Answer BHC: The point was also comment by the reviewer 2. We review all, tables and Graphs and change the size. We corrected font type and size, punctuation errors, etc. Also, we modified English text. Now the Table and Graphs are light up better. All the table and graphs were changed. Now you can see it in the new revision.

  1. A very interactive study showing the current problems of the world.

Answer BHC: Thank you for you comments. I agree with you. The correct treatment of patients with bone fragility fractures is a global public health problem, and a responsibility of of all those who care for this type of patients.

  1. Original topic, brings innovation a different perspective.

Answer BHC: Your comment is very gentle. The correct implementation of a multidisciplinary FLS with the commitment of many different specialties is a different perspective, not novel as you can see in the text, but efficient and different.

  1. The text is clear and easy to read.

Answer BHC. Thanks. However according to the reviewer 2 we did a major revision. The changes were included, and we consider that the changes have improved the article

  1. Coherent conclusions, presented very clearly 

Answer BHC. Thanks.

Reviewer 2 Report

The authors show a picture of their organizational model. This can be interesting, in order to take this FLS as an example and discuss its correctness with respect to the criteria of the IOF and the defects and errors that can be committed by anyone who wants to set up such a model. But, in order to be published, I believe that, in addition to an accurate description of the organizational model, more data are needed on the improvements and advantages that this model has brought over the years. English is partially acceptable but needs to be improved as some parts of the work are hard to understand. The manuscript should be presented in a more orderly and clearer way. Furthermore, I believe that it is incorrect to include patients who have fractured at 6 and 12 months and more than 12 months in the study.

Author Response

Answers to reviewer 2.

Thanks for your gentle review and comments. They certainly provide a critical and current point of view. After making the modifications suggested by your review, the work has gained enormously in clarity and quality. I respond to your comments.

Reviewer 2 comment: The authors show a picture of their organizational model. This can be interesting, in order to take this FLS as an example and discuss its correctness with respect to the criteria of the IOF and the defects and errors that can be committed by anyone who wants to set up such a model.

Answer BHC:  The aims of our study (Page 3 line 100) were: to describe the characteristics of the FLS and how it works. To know the type of patients being treated, with an emphasis on gender differences, and to know the degree of completion of The Best Practice Framework quality standards according to IOF. As you see, the third one is about to know the correctness with respect to the criteria of the IOF. The defects and errors that can be committed have been discussed in pages 1 and 2 of the discussion.

Reviewer 2 comment: But,  in order to be published, I believe that, in addition to an accurate description of the organizational model, more data are needed on the improvements and advantages that this model has brought over the year.

Answer BHC: I appreciate your comment. The authors consider that the detailed description of the model makes its replication possible. In addition, the data on the characteristics of the patients and the differences between sexes, and to know the fulfillment of the IOF standards and the defects and errors committed and discussed about our FLS model, are very relevant and justify their publication. Of course, data about advantages of the FLS model are highly relevant. We are now preparing a second work about the improvements and advantages of the HUVM-FLS model have meant during the first two years after its implementation. We now know our re-fracture and mortality rate before the FLS [7. Rey-Rodriguez MM, Vazquez-Gámez MA, Giner M, et al. Incidence, morbidity and mortality of hip fractures over a period of 20 years in a health area of Southern Spain. BMJ Open 2020;10:e037101. doi:10.1136/bmjopen-2020-037101]. We are doing the statistical about the re-fracture and mortality after the FLS.  We will publish these results when we finish the analysis. But these results are not the aim of this study.

Reviewer 2 comment: English is partially acceptable but needs to be improved as some parts of the work are hard to understand.

Answer BHC: An English expert have made a new revision of English, to improve it. In fact, this comment has been of great importance to improve the work.

Reviewer 2 comment: The manuscript should be presented in a more orderly and clearer way.

We did a lot of changes respect to this comment. You can see that we did a “Major Revision” . It includes title, all the text, graphs, Table, etc…. Thank for you recommendation.

Reviewer 2 comment: Furthermore, I believe that it is incorrect to include patients who have fractured at 6 and 12 months and more than 12 months in the study.

Answer BHC: I appreciate this comment as it is very intelligent and a hot topic among those of us who do OP. We included patients who had a bone fragility fracture in the previous 24 months.

This aspect was subject of discussion by all the authors in the period of creation of the FLS, before the start. Finally, we decided to take two years to give a treatment opportunity to patients with very high risk and high risk of imminent fracture. These novels concepts were based in: Johansson H, et al. Imminent risk of fracture after fracture. Osteoporos Int. 2017;28(3):775-780 and Kanis JA, et al. Characteristics of recurrent fractures. Osteoporos Int. 2018;29(8):1747-1757.

Finally, a recent article confirms this fact: Kanis JA, et al. Algorithm for the management of patients at low, high and very high risk of osteoporotic fractures. Osteoporos Int. 2020;31(1):1-12.

This was the reason for included patients with high risk and very high risk of imminent fracture. That is, within two years after the fragility fracture, in order to improve the poor outcomes particularly in those frail patients over 80 years old.

Round 2

Reviewer 2 Report

The corrections are well made, but the study shows not sufficient results to be accepted in this review.